# Sea level rise and flooding of hazardous sites in marginalized communities across the United States

Lara J. Cushing [1] ✉, Yang Ju [2], Seigi Karasaki [3], Scott Kulp[4], Nicholas Depsky[3], Alique Berberian[1], Jessie Jaeger [5], Benjamin Strauss [4] & Rachel Morello-Frosch [5,6] ✉

Sea level rise (SLR) increases the risk of flooding at coastal sites that use and produce hazardous substances. We assess whether socially marginalized populations in the United States are more likely to be impacted by projected SLR-related flooding of hazardous sites that could result in contaminant releases. We identify 5500 facilities at risk of a 1-in-100-year flood event by 2100 under a scenario of continued high greenhouse gas emissions, including coastal power plants, sewage treatment facilities, fossil fuel infrastructure, industrial facilities, and formerly used defense sites. Seven states (Louisiana, Florida, New Jersey, Texas, California, New York, and Massachusetts) account for nearly 80% of projected at-risk facilities. Controlling for population density and county, a one standard deviation increase in the proportion of linguistically isolated households, neighborhood residents identifying as Hispanic, households with incomes below twice the federal poverty line, households without a vehicle, non-voters, and renters is associated with 19-41% higher likelihood of having a site at risk of SLR-related flooding within 1 kilometer (odds ratios [95% confidence intervals]: 1.19 [1.09, 1.31], 1.22 [1.08, 1.37], 1.27 [1.16, 1.39], 1.35 [1.21-1.51], 1.36 [1.21, 1.53], and 1.41 [1.32, 1.52], respectively). Results elucidate the need for disaster planning, land-use decision-making, as well as mitigation strategies that address the inequitable hazards and potential health threats posed by SLR.

Global sea level has risen more than 11 cm over the last three decades and that rate is accelerating[1], leading to an increase in coastal flooding due to high tides, waves, storm surge, El Niño events and other factors. Extreme coastal flooding is projected to more than double by 2050 across much of the world[2]. By 2100, nearly all of the coastal United States (U.S.) is expected to experience elevated water levels on a daily basis that today occur only twice per century[3], with a rapid increase in the frequency of

high tide flooding projected to begin in multiple cities during the next decade[4].

Extreme flood events result in the release of toxic substances into the environment. For example, over 200 contaminant releases were reported in the Texas Gulf Coast after flooding resulting from Hurricane Harvey in 2017. Over 10 million pounds of regulated air pollutants were released from refineries, petrochemical, and other industrial facilities[5], and the catastrophic explosion of a chemical plant due to the

[1]Department of Environmental Health Sciences, University of California, Los Angeles, CA, USA. [2]Department of Land Resources and Tourism, School of Geography and Ocean Science, Nanjing University, Nanjing, China. [3]Energy and Resources Group, University of California, Berkeley, CA, USA. [4]Climate Central, Princeton, NJ, USA. [5]School of Public Health, University of California, Berkeley, CA, USA. [6]Department of Environmental Science, Policy and Management, University of California, Berkeley, CA, USA. ✉e-mail: lcushing@ucla.edu; rmf@berkeley.edu

loss of power for refrigeration necessitated the evacuation of 40,000 people[6]. Around the world, industrial facilities are disproportionately located along coastlines due to the historical importance of maritime trade to the establishment of industrial port cities, strategic access to global trade routes for raw materials and finished products via ports, and need for sea water for cooling and wastewater disposal. Marginalized racial and ethnic groups are more likely to live near hazardous waste sites and industrial facilities, and fenceline communities are typically subject to multiple forms of discrimination resulting in limited financial, political, and social capital to mitigate contaminant exposures[7]. Moreover, longitudinal analyses show that disproportionate hazard burdens faced by racially and economically marginalized groups are largely due to discriminatory land-use, permitting, and facility siting decisions[8–11]. Racial residential segregation and the inequitable distribution of stormwater infrastructure further contribute to racialized patterns of flood risk across U.S. cities[12].

Building upon a prior California analysis[13], we conducted a nationwide equity assessment of flood risk at hazardous sites in the U.S. due to sea level rise (SLR). We derived probabilistic estimates of flood risk in 2050 and 2100 across an expanded range of legacy contamination sites and facilities that contain, handle, produce or emit hazardous substances. We then assessed the geographic distribution of at-risk sites with respect to multiple present-day measures of social marginalization, including race/ethnicity, poverty (household income below twice the federal poverty line), voter turnout, housing tenure, and linguistic isolation. Our objectives were to characterize inequities in residential proximity to hazardous sites at risk of future flooding due to sea-level rise and identify communities where additional resources are needed to prevent exposure to toxic substances and enhance climate resilience.

## Results

### Hazardous sites at risk of flooding

We first assessed the annual probability of at least one flood exceeding the land elevation of over 47,646 coastal hazardous site locations compiled from one proprietary and four publicly available administrative data sources (Supplemental Table S1). We considered all sites within counties with land area below the 18 m elevation above current mean higher high water line across all coastal U.S. states and Puerto Rico. We defined sites as at risk if their projected annual probabilities exceeded 0.01 (i.e., they were threatened by a 1-in-100-year flood event) integrated across the full distribution of SLR projections using the law of total probability for one low (Reference Concentration Pathway [RCP] 4.5) and one high (RCP 8.5) greenhouse gas emissions scenario (see "Methods").

We found that over 11% of coastal sites in our analysis are at risk of SLR-related flooding by 2100 under the high emissions scenario (RCP 8.5) (Table 1). Figure 1 shows the distribution of at-risk sites by state or territory under RCP 8.5 in 2050 and 2100. Seven states (Louisiana, Florida, New Jersey, Texas, California, New York, and Massachusetts) account for nearly 80% of projected at-risk sites in 2100 (Fig. 1). Restricting greenhouse gas emissions to the low emissions scenario makes little difference in terms of the number of projected sites at risk in the near term (2050) but would reduce the number of at-risk sites from 5500 to 5138 (a reduction of 362 or 7% of sites) in the long term (2100) (Table 1). Oil and gas wells and industrial facilities that emit quantities of hazardous substances that require reporting to the U.S. Environmental Protection Agency's Toxic Release Inventory (hereafter "TRI sites") make up the largest proportion of sites we considered and sites at risk (Table 1). Under the high emissions scenario (RCP 8.5), over a fifth of coastal sewage treatment facilities, refineries and formerly used defense sites, roughly a third of power plants, and over 40% fossil fuel ports and terminals are projected to be at risk by 2100 (Table 1).

### Affected communities

We next considered the distribution of at-risk sites with respect to community demographics and indicators of social marginalization derived from three secondary datasets: the American Community Survey, a proprietary data source on recent voter turnout, and the federal Climate and Economic Justice Screening Tool[14]. We utilized census block groups as the geographic unit of analysis (hereafter "neighborhoods") and considered block groups with at least one at-risk site located within 1 km of a populated area as being potentially affected (see "Methods"). Given the prominence of racial discrimination as a means of establishing and maintaining social inequality in the U.S.[15], we considered measures of racial and ethnic makeup, as well as indicators of socioeconomic status, civic engagement (voter turnout), and vulnerability that relate to communities' ability to anticipate, mitigate, and cope with flooding, such as age, linguistic isolation (% of households where no one 14 years or older speaks English "very well"), and vehicle ownership.

Table 2 summarizes the population characteristics of neighborhoods near versus far from hazardous sites at risk of flooding due to SLR in 2100 under a high emissions scenario. Figure 2 shows the increase in the likelihood of an at-risk site within 1 km per one standard deviation increase in each demographic and social vulnerability measure, which we estimated using logistic regression models controlling for population density and county to minimize bias related to the higher concentration of people of color and renters in urban areas and demographic variation across U.S. regions.

Compared to other coastal neighborhoods, neighborhoods with one or more at-risk site nearby have lower voter turnout, proportions of residents identifying as Asian/Pacific Islander, and individuals under the age of 18, and higher present-day proportions of renters, households living in poverty, residents identifying as Hispanic and Black, linguistically isolated households, households without a vehicle, single-parent households, and individuals over the age of 65 (Table 2). In the multivariable regression models, all these bivariate associations remained statistically significant with the exception of the proportion of Black and Asian/Pacific Islander residents (Fig. 2). Neighborhoods designated as disadvantaged by the federal Climate and Economic Justice Screening Tool, a nationwide composite assessment of cumulative impact associated with multiple measures of social vulnerability (e.g., poverty) and the presence of climatic and environmental hazardous, had a 50% higher odds of having an at-risk site within 1 km, compared to other coastal, non-disadvantaged neighborhoods (odds ratio [OR] and 95% confidence interval [CI] = 1.50 [1.23, 1.83], Fig. 2). A one standard-deviation increase in the proportion of residents over age 65, linguistically isolated households, residents identifying as Hispanic, households in poverty, households without a vehicle, non-voters, and renters was associated with 15–41% higher likelihood of an at-risk site within 1 km (ORs and 95% CIs shown in Fig. 2). Associations were similar when we considered the presence of at-risk sites within 3 km instead of 1 km of neighborhoods (Supplementary Table S2).

Among neighborhoods within 1 km of an at-risk site, social marginalization was also associated with an increase in the number of at-risk sites nearby and the severity of flood risk across those sites (Supplementary Fig. S1). Here we quantify flood risk severity by estimating the neighborhood expected annual exposure (EAE), calculated by summing the annual probabilities of at least one flood occurring at all hazardous sites within 1 km of populated portions of census block groups. This expected value reflects the total number of sites likely to be exposed to flooding in a given year—either 2050 or 2100 (see "Methods"). Among neighborhoods with an at-risk site within 1 km, a one standard deviation increase in the proportion of Hispanic residents, households in poverty, households without a vehicle, non-voters, and renters was associated with a 7–13% higher number of at-risk sites in 2100 under RCP 8.5 and a 0.10–0.21 unit increase in EAE

**Table 1 | Number and type of hazardous sites at risk of sea level rise-related flooding by greenhouse gas emission scenario and year across the coastal U.S**

| Category | Total number of facilities in analysis | Number (%) at risk, RCP 4.5 | | Number (%) at risk, RCP 8.5 | |
|---|---|---|---|---|---|
| | | 2050 | 2100 | 2050 | 2100 |
| Power plants (nuclear & fossil fuel) | 443 | 84 (19.0) | 125 (28.2) | 85 (19.2) | 134 (30.2) |
| Animal operations | 1148 | 87 (7.6) | 111 (9.7) | 88 (7.7) | 115 (10.0) |
| Sewage treatment facilities | 2582 | 379 (14.7) | 525 (20.3) | 384 (14.9) | 564 (21.8) |
| Hazardous waste treatment & disposal | 515 | 44 (8.5) | 68 (13.2) | 46 (8.9) | 74 (14.4) |
| Other industrial facilities (Toxic Release Inventory) | 15,222 | 1049 (6.9) | 1679 (11.0) | 1073 (7.0) | 1870 (12.3) |
| Solid waste landfills & incinerators | 948 | 50 (5.3) | 79 (8.3) | 51 (5.4) | 90 (9.5) |
| Cleanup sites & sites with radioactive material | 604 | 64 (10.6) | 100 (16.6) | 66 (10.9) | 111 (18.4) |
| Refineries | 67 | 9 (13.4) | 14 (20.9) | 9 (13.4) | 16 (23.9) |
| Fossil fuel ports and terminals | 663 | 196 (29.6) | 275 (41.5) | 199 (30.0) | 293 (44.2) |
| Active oil & gas wells | 24,095 | 1592 (6.6) | 1895 (7.9) | 1597 (6.6) | 1944 (8.1) |
| Formerly used defense sites | 1359 | 186 (13.7) | 267 (19.6) | 190 (14.0) | 289 (21.3) |
| Total | 47,646 | 3740 (7.8) | 5138 (10.8) | 3788 (8.0) | 5500 (11.5) |

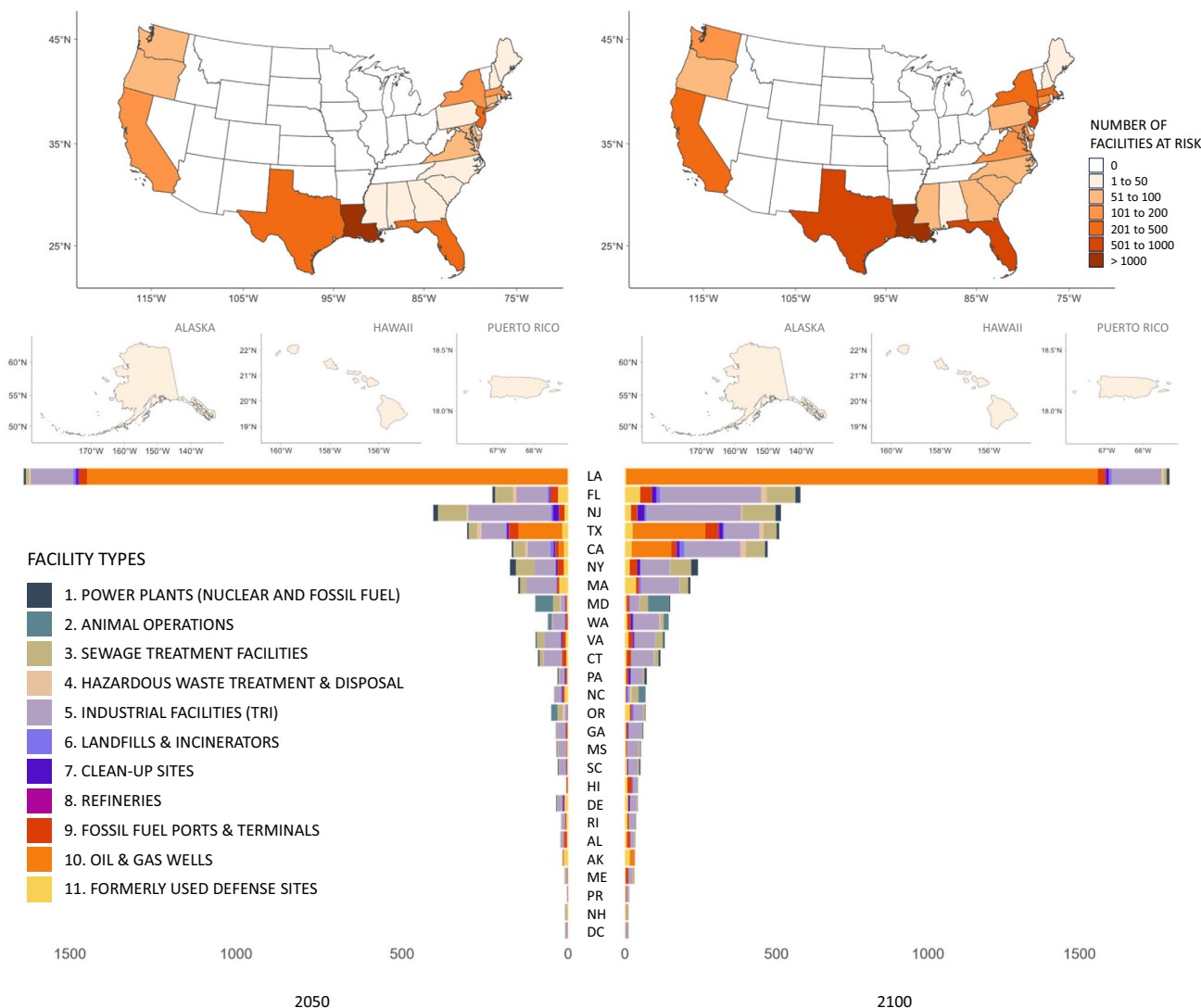

**Fig. 1 | Number of sites at risk of flooding due to sea level rise in (left) 2050 and (right) 2100 under a high emissions scenario (RCP 8.5) by state and type.** States are shaded by the total number of at-risk sites, with darker colors representing a higher number of sites at risk (maps). The number of sites at risk in each state is broken down by type, with each facility type represented by a unique color (bar chart).

**Table 2 | Characteristics of coastal block groups (n = 51,772) with and without at-risk sites within 1 km of populated areas in 2100 under RCP 8.5 across the U.S**

| | No at-risk sites (n = 40,233) Median [25th,75th percentile] | One or more at-risk sites (n = 11,539) Median [25th,75th percentile] | P-values |
|---|---|---|---|
| % non-voters | 26.2 [19.6, 35.1] | 29.1 [21.9, 37.9] | 0.00 |
| % poverty | 25.6 [13.1, 43.6] | 29.3 [15.5, 47.9] | 0.00 |
| % renters | 34.6 [16.2, 60.7] | 48.3 [24.3, 72.6] | 0.00 |
| % racial and ethnic minorities | 44.8 [19.8, 80.3] | 47.2 [21.1, 80.5] | 0.01 |
| % Hispanic | 11.3 [3.7, 29.4] | 12.6 [4.2, 33.8] | 0.00 |
| % Black | 4.3 [0.0, 19.3] | 5.3 [0.5, 21.7] | 0.00 |
| % Asian & Pacific Islander | 2.5 [0.0, 9.8] | 2.3 [0.0, 9.2] | 0.01 |
| % Native American | 0.0 [0.0, 0.0] | 0.0 [0.0, 0.0] | 0.13 |
| % other people of color | 1.6 [0.0, 4.3] | 1.5 [0.0, 4.1] | 0.04 |
| % linguistic isolation | 2.6 [0.0, 9.8] | 3.6 [0.0, 12.6] | 0.00 |
| % without a vehicle | 5.6 [1.4, 16.8] | 10.4 [2.9, 29.5] | 0.00 |
| % single parent household | 16.6 [9.1, 27.0] | 16.9 [8.9, 28.2] | 0.03 |
| % over 65 | 23.5 [13.0, 37.4] | 26.2 [14.7, 42.5] | 0.00 |
| % under 18 | 20.1 [14.3, 25.9] | 19.3 [12.7, 25.5] | 0.00 |

P-values are from a two-sided Mann–Whitney U-test of the null hypothesis that the distributions are equal. No adjustments were made for multiple comparisons.
N is slightly lower for some individual indicators due to missing data.

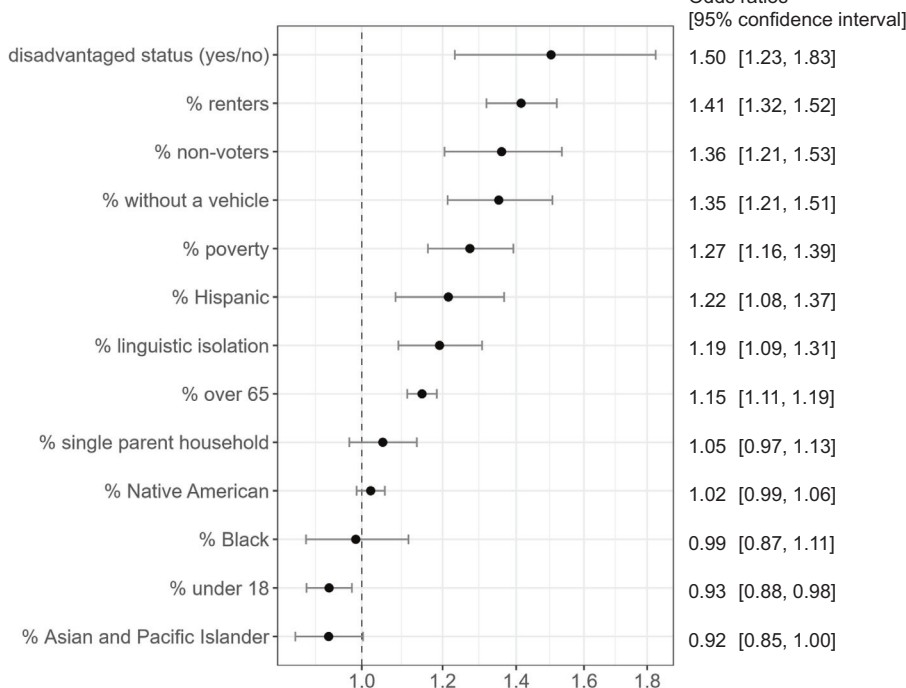

**Fig. 2 | Odds ratios and 95% confidence intervals for the association between the presence of socially marginalized groups and the likelihood of an at-risk site within 1 km under RCP 8.5, 2100 among coastal neighborhoods (N = 51,957 block groups).** Black circles are adjusted odds ratios from models that considered one population characteristic at a time and controlled for population density and county fixed effects. Error bars indicate 95% confidence intervals and were calculated using robust standard errors. The dashed line indicates no association. Disadvantaged status (as defined by the federal Climate and Economic Justice Screening Tool [CEJST]) is a binary variable; all other variables are continuous and were scaled by unit standard deviation to facilitate comparisons between effect estimates.

(see Incidence Rate Ratios, mean differences and corresponding 95% CIs in Supplemental Fig S1). These estimates again control for population density and county to minimize bias.

## Most inequitably distributed sites
Figure 3 presents concentration indices and 95% confidence intervals summarizing the degree of inequality in the distribution of at-risk sites with respect to demographic and social marginalization indicators. We utilized concentration indices to identify the categories of facilities that were the most inequitably distributed for particular populations. Similar to the Gini coefficient commonly used to characterize income inequality, a concentration index ranges from −1 to 1, with negative values (in orange) indicating that the burden of at-risk sites is disproportionately higher for more marginalized groups and positive

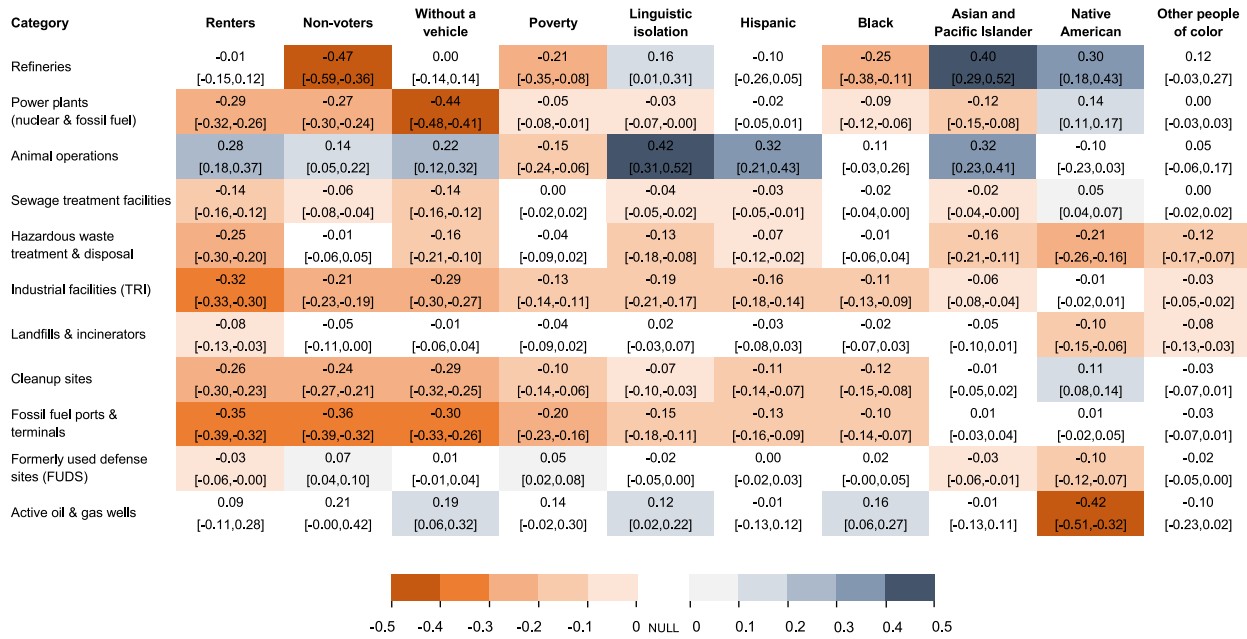

**Fig. 3 | Concentration indices and 95% confidence intervals for the cumulative distribution of at-risk facilities with respect to selected demographic and social marginalization measures under RCP 8.5, 2100.** Negative values (in orange) indicate a disproportionately higher burden of at-risk sites for marginalized groups, while positive values (in blue) indicate that the burden is disproportionately lower for these groups. White values indicate a lack of statistical significance at $P > 0.05$. No adjustments were made for multiple comparisons.

values (in blue) indicating that the burden is disproportionately lower for marginalized groups. Values are shaded white when confidence intervals include the null, indicating no significant evidence of a disparity. Concentration curves corresponding to the indices given in Fig. 3 are included in Fig. 4. These display the distribution of at-risk sites with respect to demographic and social marginalization measures, with the area between the curve and diagonal line of equality being equivalent to the concentration index (e.g. between −1 and 1). To increase the legibility of Fig. 4, for each demographic and social marginalization measure we display only the five site categories with the strongest concentration indices, while the full set of concentration indices is shown in Fig. 3.

At-risk power plants, industrial TRI sites, clean-up sites, and fossil fuel ports and terminals disproportionately burdened neighborhoods with higher proportions of renters, non-voters, households without a vehicle, households living in poverty, and linguistically isolated households, as indicated by negative concentration index values in Fig. 3 and curves above the line of equality in Fig. 4. In contrast, at-risk concentrated animal feeding operations and active oil and gas wells more often did not disproportionately burden marginalized groups, as indicated by mostly positive concentration index values in Fig. 3 and curves below the line of equality for many panels in Fig. 4, although there were exceptions. At-risk refineries disproportionately burden neighborhoods with higher proportions of non-voters, households in poverty, and Black residents, and at-risk TRI facilities disproportionately burden neighborhoods with higher proportions of Black, Hispanic and Asian/Pacific Islander residents. In contrast, neighborhoods with a higher proportion of Native American residents are projected to be disproportionately burdened by at-risk active oil and gas wells, hazardous waste sites, landfills, and formerly used defense sites (Figs. 3, 4).

Conclusions about which at-risk site types are inequitably distributed are largely but not entirely consistent across different metrics of flood risk (number of at-risk sites, which is presented in Fig. 3 vs. EAE across sites which is presented in Supplementary Fig S2). For example,

neighborhoods with higher proportions of renters, linguistically isolated households, and households without a vehicle were not burdened by a disproportionate share of at-risk refineries (Fig. 3), but when assessing EAE, they were disproportionately burdened (Supplementary Fig S2 and S3). Similarly, Hispanic and Asian/Pacific Islanders were not burdened by a disproportionate share of at-risk refineries (Fig. 3), but are disproportionately burdened when considering EAE (Supplementary Fig S2 and S3). This may be because although the number of at-risk refineries tends to be higher near neighborhoods with smaller proportions of these residents, the severity of projected flooding at those refineries is higher than it is near other neighborhoods.

## Discussion

We present a national assessment of projected SLR-related flooding threats to multiple categories of coastal sites and facilities that contain, use or produce hazardous materials. Our results show that of the more than 47,600 coastal facilities in the U.S. included in our analysis, over 11% (5500 facilities) are projected to be at risk of a 1-in-100-year or more frequent flood event by the end of the 21st century (2100) under a high (RCP 8.5) greenhouse gas emission scenario. A handful of states, including Louisiana, Florida, New Jersey, Texas, California, New York, and Massachusetts account for nearly 80% of projected at-risk sites.

Facilities at risk include 22% of coastal sewage treatment facilities, 24% of refineries, 44% of fossil fuel ports and terminals, 12% of industrial facilities, 21% of formerly used defense sites and 30% of fossil fuel and nuclear power plants. A prior study estimated the number of wastewater treatment plants and service populations across the U.S. that could be exposed to SLR scenarios from 1 to 6 ft, with projections ranging from 60 impacted treatment plants serving 4 million people to 394 plants serving over 31 million people[16]. That analysis did not incorporate elevated water levels due to tides, waves, and storm surge, which likely explains why we projected a larger number of sewage treatment facilities to be at risk of SLR-related flooding in the RCP 8.5 scenario. Another prior assessment of how unmitigated

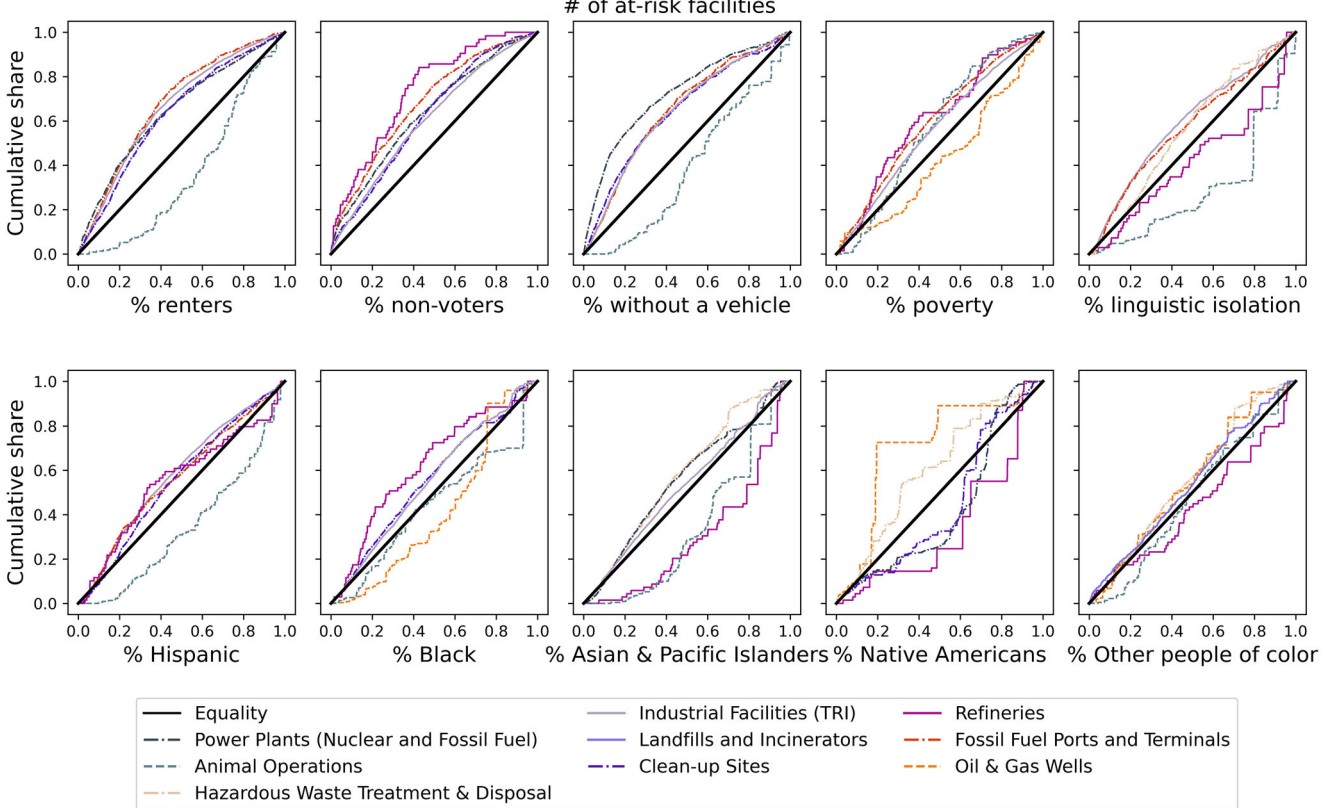

**Fig. 4 | Cumulative distribution of the number of at-risk sites with respect to selected demographic and social marginalization measures under RCP 8.5, 2100.** The X-axis gives the cumulative share of block groups in descending order of each of the demographic and social marginalization variables. Curves above the equality line indicate a disproportionately higher burden of at-risk sites for marginalized groups, while a curve below the equality line indicate that the burden is disproportionately lower for these groups. For example, the top left panel shows that the 50% of low-lying block groups with the highest proportion of renters (indicated by the x-axis value of 0.5) host roughly 80% of at-risk fossil fuel ports and terminals and industrial facilities (indicated by y-axis values of 0.8), whereas if these sites were equitably distributed, the y-axis value would be close to 0.5 and the curves would fall closer to the bolded diagonal line of equality. For legibility, the top 5 facility categories with the strongest concentration indices for each measure are shown.

greenhouse gas emissions could affect U.S. power-generating capacity in 2100 among power plants in coastal areas estimated a similar number of power plants at risk as we did in our analysis. That study additionally considered the generation capacity of at-risk power plants. The authors found significant variation across states with exposed power capacities relative to current generation capacities being highest in Delaware, New Jersey and Florida (80%, 63% and 43%, respectively)[17].

Our analysis shows that industrial facilities that are part of the Toxic Release Inventory make up nearly a third (34%) of the total sites at risk of SLR-related flooding (N = 1870), second to fossil fuel infrastructure (41%), including refineries, fossil fuel ports and terminals and active oil and gas wells. Because we did not include pipelines in our analysis, our projections of the extent to which the nation's fossil fuel infrastructure may threaten coastal communities due to SLR-related flooding and associated contaminant releases are likely an underestimate. Indeed, extreme weather events such as Hurricanes Katrina, Rita and Harvey, while very different from the slower moving, incremental flooding related to SLR, have dramatically revealed the vulnerability of industrial facilities and oil and gas infrastructure. Flooding following these hurricanes led to oil and chemical spills, pipeline ruptures, as well as excess air pollutant emissions due to incidental releases as well as intentional shutdowns, flaring, and subsequent restarting of operations at petrochemical facilities[18–23]. Our prior equity analysis of contaminant releases related to Hurricanes Harvey, Rita and Ike found that these natural-technological (natech) disasters

disproportionately impacted Hispanic, renter, low-income, and rural populations[5]. Similarly, results in this study show significant inequities in projected SLR flooding threats to potentially hazardous facilities; communities defined as disadvantaged by the federal Climate and Economic Justice Screening Tool (CEJST) have a 50% higher odds of having an at-risk site within 1 km, and a one standard deviation increase in the proportion of linguistically isolated households, neighborhood residents identifying as Hispanic, households in poverty, households without a vehicle, non-voters, and renters is associated with 19–41% higher likelihood of an at-risk site.

Our findings align with prior equity studies of current and projected distributional burdens of flood risk among diverse populations in the U.S. A national study using Federal Emergency Management Agency maps from 2001–2019 in urban areas along with National Land Cover Data and county-level Census data found that 100-year flood zones, particularly in coastal counties, are often occupied by a higher proportion of disadvantaged populations[24]. Another study of coastal and inland areas estimated an average increase of 26.4% (24.1–29.1%) in climate change related flooding by 2050 under an RCP4.5 scenario, with the future increase in flooding risk concentrated on the Atlantic and Gulf coasts and disproportionately affecting Black communities[25]; although this study examined flooding and economic losses related to residential and non-residential properties, it did not consider risks to potentially hazardous sites. Other studies have examined flooding threats to active and legacy sites containing hazardous material. For example, a

report found low-income communities were disproportionately represented among the populations living in proximity to clean-up sites (listed or candidate sites for the Superfund program) at risk of coastal flooding under low, medium, and high SLR scenarios in the East and Gulf Coasts[26]. A follow-up study identified coastal land below 10 m of elevation as potentially exposed to rising groundwater and identified 326 Superfund sites in these coastal areas that could experience mobilization of toxic compounds from contaminated soil due to groundwater inundation driven by SLR; results also showed that socially marginalized groups in several states would be disproportionately affected by this groundwater rise scenario[27]. Another analysis of former hazardous manufacturing facilities in six U.S. cities identified more than 6000 relic industrial sites with elevated flood risk over the next 30 years (2050), with socially vulnerable groups, including people of color and low income, disproportionately likely to live in these areas[28]. Studies outside of the U.S., for example in coastal regions in India, Copenhagen, Vietnam and Italy have investigated the risks posed by climate change-driven SLR and storm surge on infrastructure and vulnerable sites[29–32], but none to our knowledge have evaluated these risks using an environmental justice framework.

Strengths of our study include the use of tax parcel data to characterize the extent of facility boundaries, a probabilistic approach to estimating SLR-related flood risk, and the application of dasymetric mapping techniques to estimate populations and community demographics near at-risk sites. Limitations of our analysis include the fact that our flood models assume that the frequency and magnitude of flood events will remain static over the next century. However, studies indicate that tropical cyclone activity is likely to intensify due to the acceleration of climate change[33–35], which would result in more damaging impacts to coastal communities[36]. Additionally, our estimations of annual probabilities of flood level exceedance, based on a modified "bathtub" approach, do not consider scenarios of groundwater intrusion and upwelling or nonlinear interactions between extreme flood events and local topography. These dynamics could cause increased flood levels at inland locations, especially where marshlands shrink, and land becomes more developed[37]. Our analysis also does not account for floodwater level attenuation particularly in areas where land is wide and flat, which may overestimate exposure during extreme storm events[38]. Locational errors for hazardous sites may have also led to over- or under-estimates of the number of at-risk sites, and data limitations precluded inclusion of other facility types, including underground storage tanks, brownfields, and non-National Priority List Superfund sites that could experience contaminant releases due to SLR-related flooding. Inaccuracies in the delineation of coastline boundaries may have resulted in the inclusion of offshore drilling sites and the overestimation of flood risk. Finally, we did not account for future flood risk mitigation efforts or population and demographic shifts, given the high degree of uncertainty in predicting these scenarios. Therefore, future actions to mitigate flood risk near hazardous sites, gentrification, displacement, migration, and other factors could change the associations we observed between demographics, measures of social marginalization, and proximity to at-risk sites.

Our analysis highlights the disproportionate burden of projected SLR-related flooding threats to hazardous sites on marginalized racial and socioeconomic groups and elevates the importance of centering environmental justice in future climate change adaptation and land-use planning strategies to protect vulnerable coastal communities from natech disasters. Given that nearly 80% of projected at-risk sites are in seven states, future in-depth work can target these areas and more precisely characterize the potential hazards posed by these facilities to nearby communities with the goal of mitigating and preventing future harmful exposures and health risks. With over 30% of nuclear and fossil fuel power plants, 23% of refineries, and 44% of fossil fuel ports and terminals in coastal areas projected to be at risk, federal reporting requirements for these facilities could be expanded to include the forecasting of SLR-related flooding threats and preventive plans for mitigation, including future relocation, to avoid catastrophic contamination. Critical to these efforts will be ensuring that federal and state agencies provide publicly available, accessible, and continually updated data on projections of SLR-related flooding threats to hazardous sites for diverse end-users, in particular at-risk communities, planners, regulatory agencies, scientists, and decision-makers[39]. Future research focusing on a smaller subset of facilities and more localized regions or municipalities could further elucidate and potentially untangle the extent to which place-based trends in industrial, economic, labor market, and housing development trajectories, demographic churning, changes in land-use decision-making as well as other shifting structural factors account for the origins and persistence of inequities in exposures to at-risk sites that disproportionately affect marginalized populations.

Finally, many other climate-related phenomena, such as groundwater rise, wildfires, landslides, major storms, and extreme heat, also threaten clean-up sites and active facilities that use and store hazardous materials[27,40,41]. To achieve a fuller picture, information on these threats should be integrated with projected SLR flood risk data. Risks may be reduced through enhanced regulatory requirements (1) for at-risk facilities to mitigate and prevent contamination threats and (2) for more robust assessment of clean-up sites to inform abatement activities and decisions about future land reuse. Action-oriented partnerships between communities living near at-risk sites and government agencies at local, state, and federal levels may increase the chances for success of these strategies by marshalling much-needed resources aimed at preventing contamination from acute natech disasters and slower-moving threats, including SLR-related flooding.

## Methods

Our analytic approach entailed four steps: 1) the identification of coastal hazardous site locations and the cleaning of associated descriptive data; 2) the estimation of future flood risk due to sea level rise at each site location; 3) the compilation of measures of demographics and social marginalization; and 4) a neighborhood-level analysis of the relationship between these measures and residential proximity to at-risk sites. We co-developed these methods with an advisory committee comprised of staff members from environmental justice and public health organizations with whom we collectively decided on greenhouse gas emissions scenarios, timeframes (2050 and 2100), flood risk metrics, categorization of sites, and the demographic and social vulnerability metrics to include[13].

### Hazardous sites

The spatial extent of our analysis was U.S counties and county equivalents with any land area below 18 meters elevation above current mean higher high water line across all coastal U.S. states and Puerto Rico (see Supplementary Fig S4). Areas farther inland are at no conceivable risk of flooding due to sea level rise this century and were therefore excluded. We scaled up an approach for a prior analysis of California[13] to compile a national dataset of active industrial facilities and other potentially hazardous sites. To achieve this, we sourced data from the U.S. Environmental Protection Agency's (EPA) Facility Registry Service (FRS)[42], the U.S. Energy Information Administration's (EIA) Energy Atlas[43] (petroleum refineries and terminals), the U.S. Army Corp of Engineers' (USACE) Waterborne Commerce Statistics Center[44] (petroleum ports) and Formerly Used Defense Sites database[45], and a proprietary dataset of active oil and gas production and stimulation wells from Enverus[TM] [46]. For the FRS, we chose to exclude remediated and closed facilities and facilities with inaccurate or imprecise locational information (e.g. latitude and longitude values derived from zip

codes only or with inaccuracy >50 m). This included sites with environmental interest "end dates" indicating they would no longer be regulated after 2020 or where records indicated contamination had been addressed or the site was permanently closed. We retained inactive facilities and facilities with expired permits because residual hazardous materials may remain at these sites. We organized the remaining sites into one of seven categories using (1) environmental permits or regulatory programs; (2) the North American Industry Classification System (NAICS) code; and/or (3) keyword filters (see Cushing et al. [13] for further detail). We made sure that each category was mutually exclusive and without duplicate entries, as sites can have more than one environmental permit and/or NAICS code and appear in more than one database. We manually removed FRS entries for refineries using refinery names and coordinates from the EIA Energy Atlas dataset.

For oil and gas wells, we utilized latitude and longitude point locations to represent sites due to the small size of well pads compared to other site categories. We identified offshore oil and gas wells as those that were beyond the boundaries of 2010 Census block groups, and excluded them from the analysis to focus on wells located on land. Block group boundaries from the National Historical Geographic Information System do not include coastal water areas and terminate at the coastline. All other site types were represented as polygons in our analysis to better approximate a site's extent. For FRS, EIA, and USACE petroleum port sites, we used the Google API[47] to (re)geo-code addresses, then joined the resulting coordinates to tax parcels obtained from Loveland Technologies (now Regrid)[48] to approximate their spatial boundaries. Wherever a site's geocoded location overlapped with a tax parcel, we used that parcel to approximate that site's spatial extent for the purposes of projecting flood risk. Around 79% of our geocoded site locations fell within tax parcel boundaries. For sites that did not intersect tax parcels, we approximated boundaries by drawing a buffer equal to the median parcel area of intersected parcels for each site category (See Supplementary Material Table S1 for median areas applied for each category). For formerly used defense sites (FUDS), spatial data were available in both point and polygon format. Not all sites had a point or polygon, while some sites had both. To ensure we included all FUDS in our dataset, we (i) first included all facilities with polygon data, then (ii) identified facilities with point locations falling outside of these provided polygon boundaries, and (iii) drew a buffer around these point facilities using a circular radius that would result in the median area observed among facilities in (i). Wherever a polygon boundary overlapped with a buffered point, we clipped the latter based on the physical extent of the former ($n = 25$). For overlaps between two polygon boundaries, we used an overlap ratio (calculated as the area of the overlap divided by the area of the smaller polygon) to determine whether to split the overlapping area evenly between two facilities with minimal overlap (≤45%, $n = 33$), or to merge substantially overlapping facilities together into one (>45% overlap, $n = 42$). We resolved 13 complex cases involving three or more overlapping facilities manually on a case-by-case basis. The result of this process was a dataset of FUDS boundaries derived from original point locations or polygon boundary extents that contained no spatial overlaps between sites. For all sites, we then clipped parcels and circular buffer areas at the coast if they extended past the mean high tide line.

As a final cleaning step, we flagged and subsequently consolidated duplicate sites ($n = 656$) if they met three criteria: they were assigned to the same category, had identical geo-coded coordinates, and were associated with the same or similar addresses (we quantified similarity using a fuzzy text match). We retained facilities with identical coordinates and similar addresses if they had been assigned to different categories ($n = 232$). We dropped facilities with matching coordinates but dissimilar addresses ($n = 30$) if their geocoded

coordinates appeared inaccurate or implausible via manual visual inspection (e.g., located in the middle of a forest far away from any established roads).

## Flood risk projections

We used the same approach to assess flood risk at individual site locations as detailed in Cushing et al. (see Supplementary Fig S5)[13,49,50]. In brief, we considered probabilistic sea level rise projections[51] for two greenhouse gas emissions scenarios (Reference Concentration Pathway [RCP] 4.5 and 8.5) and 2 years (2050, 2100)[51]. For each site, year, and emissions scenario, we estimated the total annual probability of at least one flood event exceeding, in height, the 25th percentile of land elevation for a given site's parcel or buffer boundary. Projections account for vertical land movement and coastal flood height return level curves using methods from Tebaldi et al. and updated tide station data from across the United States[52]. We derived elevation profiles from NOAA's Coastal Topographic Lidar digital elevation model[53], and estimated the annual flooding probabilities using Equation (1) from Buchanan et al.[50] We considered sites to be at risk if their projected annual probabilities exceeded 0.01 (i.e., threatened by a 1-in-100 year flood event). We also summed these probabilities across sites to derive a total expected annual exposure (EAE) across all at-risk sites within a given distance of neighborhood (block group) boundaries.

## Neighborhood demographics and social marginalization

We used 2010 U.S. Census block group boundaries as our definition of geographic neighborhoods. Census block groups are generally contiguous geographic areas that contain between 600 and 3000 people and are the smallest unit for which the U.S. Census Bureau reports a full range of demographic statistics. We used the U.S. Census American Community Survey's (ACS) 2015–2019 five-year estimates[54] to approximate demographic characteristics at the block group level: age (% under 18 and % 65 and older), race/ethnicity (% people of color, defined as the inverse of % non-Hispanic White and disaggregated into % Hispanic, and % non-Hispanic [NH] Black, NH Asian or Pacific Islander, NH Native American, and NH other including multiracial), poverty (% below twice the federal poverty line), housing tenure (% renter-occupied units), vehicle ownership (% of households without a vehicle), family structure (% single parent-headed households), linguistic isolation (% of households where no one 14 years or older speaks English "very well"). We used voter turnout data from the 2016 and 2020 general elections from Catalist's National Database to approximate civic engagement (% of registered voters that did not vote averaged across the two elections). Finally, we used the federal Climate and Economic Justice Screening Tool (CEJST) that identifies disadvantaged communities in all 50 states, the District of Columbia, and U.S. Territories[55]. Developed as part of the Justice40 Initiative, CEJST was used by federal agencies to identify disadvantaged communities facing disproportionate climate and environmental burdens as well as economic marginalization. CEJST identifies disadvantaged communities through eight categories of vulnerability metrics related to climate change, energy, health, housing, legacy pollution, transportation, water and wastewater, and workforce development. Census tracts are identified as disadvantaged if they meet 90th percentile thresholds for indicators within any of the eight categories and are at or above the 65th percentile for low-income.

## Statistical analysis

We began by identifying and including only counties in our study area with at least one site at risk under RCP 8.5 by 2100. We then further restricted the geographic extent of our analysis to "coastal" block groups in these counties within 3-km Euclidean distance of the 10-m elevation line above mean higher high water line. Our primary outcome of interest was the presence of at least one at-risk site within 1 km. We considered block groups to have this outcome if they

contained populated areas within a kilometer of at least one at-risk site (see Supplementary Fig S4). Because block groups can be quite large in rural areas, we utilized gridded population estimates at a 30 × 30 m resolution[56] to define populated portions of block groups with the exception of Alaska, Hawaii and Puerto Rico for which these estimates were not available and where we therefore relied on block group boundaries alone. We secondarily considered (1) the total number of at-risk sites within 1 km, and (2) the sum of annual flood event probabilities (total "expected annual exposure", EAE) across all at-risk sites within 1 km. We conducted sensitivity analyses considering alternate versions of these outcomes using a 3 km rather than 1 km buffer distance.

We examined descriptive statistics and correlation coefficients between our demographic measures and indicators of social marginalization and compared the distribution of neighborhood characteristics between exposed and unexposed block groups using Mann-Whitney $U$ test because variables were not normally distributed. We then ran multivariable regression models for each outcome variable and vulnerability indicator pair, with block-group population density (people per square kilometer), and county fixed effects as additional independent variables. We did not include multiple demographic or social marginalization indicators in the same model due to multicollinearity. We chose to include population density as a potential confounder due to known associations between race/ethnicity, population density, and proximity to industrial facilities[57,58]. We included county fixed effects to control for regional demographic differences and in effect compare block groups with and without at-risk sites within the same county. We scaled continuous variables by unit standard deviation (SD), using the mean and SD from all block groups in our universe to allow for easier comparisons between effect estimates. In our primary analysis, we used a logistic model to estimate the odds of proximity to an at-risk site (yes/no variable). Restricting to exposed block groups, we used a negative binomial model to estimate associations with the number of sites nearby (count variable) and a linear model to estimate associations with EAE (continuous variable). We used county-clustered robust standard errors to control for the spatial autocorrelation.

Finally, we used concentration curves to visualize the cumulative distribution of the number of exposed facilities and EAE with respect to each indicator of social marginalization. We also derived the concentration index ($C$) equal to the area beneath the curve and line of equality in our concentration plots to quantify the cumulative distributions. $C$ ranges between −1 and 1, with negative values indicating that block groups with higher proportions of residents from socially marginalized groups have a greater number of at-risk facilities and EAE, and positive values indicating they have a smaller burden of at-risk sites and EAE. A $C$ value with a confidence interval that includes the null value of 0 indicates that the number of exposed facilities and EAE are similar between more and less marginalized populations. We calculated C using all coastal block groups, and we calculated separate indices for each facility category focusing on year 2100 under RCP 8.5. Concentration curves and indices were computed using R (version 4.5.0). All other statistical analyses were conducted using Python (version 3.13.7).

### Reporting summary
Further information on research design is available in the Nature Portfolio Reporting Summary linked to this article.

## Data availability
The datasets generated during the current study are available from the Toxic Tides maps in Climate Central's Coastal Risk Screening Tool (flood risk projections, https://coastal.climatecentral.org/), GitHub (analytic dataset and code, https://github.com/yangju-90/toxic_tides_us), and Zenodo (analytic dataset and code, https://doi.org/10.5281/zenodo.16925499).

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

## Acknowledgements

This project has been funded wholly or in part by the United States Environmental Protection Agency (EPA) under assistance agreement 84003901 to L.J.C. The contents of this document do not necessarily reflect the views and policies of the EPA, nor does the EPA endorse trade names or recommend the use of commercial products mentioned in this document. We thank the Toxic Tides Advisory Council—comprised of community leaders from the Asian Pacific Environmental Network, Central Coast Alliance for a Sustainable Economy, Physicians for Social Responsibility Los Angeles, Public Health Institute, and WE ACT for Environmental Justice—for advising on the methods.

## Author contributions

L.J.C. contributed to manuscript writing and jointly conceived of the project, acquired the funding, and supervised the work. Y.J. conducted the statistical analysis and reviewed and edited the manuscript. S Karasaki contributed to data curation and manuscript writing, and prepared figures and tables. S.Kulp conducted the flood risk projections and reviewed and edited the manuscript. N.D. and J.J. contributed to data curation. A.B. contributed to data curation, preparation of figures and tables, and edited the manuscript. B.S. jointly conceived of the project and acquired the funding, and reviewed and edited the manuscript. R.M.F. contributed to manuscript writing and jointly conceived of the project, acquired the funding, and supervised the work.

## Competing interests

The authors declare no competing interests.
