## [Transparent Peer Review file · Nature Communications]

Environmental justice, sea level rise and flooding of hazardous sites across the United States

Corresponding Author: Dr Lara Cushing

Version 0:

Reviewer comments:

Reviewer #1

(Remarks to the Author)

Dear Authors,

This is an interesting paper and I appreciate the hard work you put into to get to this point. You have provide a rigorous analysis of hazardous sites at risk from future flooding caused by sea level rise. They focus on the types of communities impacted by these exposures and bring to light the significant disparities. The data and methods for this paper are technically sound and well documented. They make some excellent points and compare their work with the current state of the literature. They make some very good points, some of which I note below. In my opinion, there are also some shortcomings that should be addressed to allow the paper to move the field forward.

This paper focuses exclusively on the United States. It is very well done and leverages important and relevant data sets. It is also timely and relevant as others have worked to call attention to this threat. However, it does not connect its findings and work to a global audience. I think the paper would be stronger if there was a thread that connected the US context with other countries.

The methods are predominantly clear, articulate, and rigorous. The major issue I have with the methods is that there are no spatial visualizations in the supplementary material to help people understand their process. One example is their use of Euclidean distance. It's my understanding that they measured 3 kilometers towards shore from the 10 meter above mean high tide line to find low-lying areas. If that's true that method has flaws since those locations could be higher in elevation depending on the local topography. Maps explaining these methods would greatly improve their methods section.

Another concern was their use to CJEST. The authors mentioned it in the discussion and in their figure 2. However there is no mention of it in their methods. This dataset should be more clearly defined and explained. Since the dataset is a combination of the other variables they are using it should be more thoroughly scrutinized and then explained.

Their results are a meaningful contribution to the field and bring to light the significant risk to vulnerable populations from hazardous waste sites at risk of sea level rise. The data they present supports this conclusion. Their findings that 70% of the sites are in seven states also gives policy makers and advocates an area of focus. The subsequent recommendation of making sure information of these risks is available seems narrow.

The results from this paper are interesting and possibly significant. It would help if they further contextualized their findings relative to other hazards. Is sea-level rise the greatest threat to hazardous waste sites? Looking at the Summers, K., Lamper, A., & Buck, K. (2021) paper would help to support their work.

(Remarks on code availability)

Reviewer #2

(Remarks to the Author)

The paper focuses on a very important topic and attempts to analyze the extent of exposure of SLR-related flooding of hazardous sites to the socially marginalized people in the USA.

The authors focus on seven states in the USA (Florida, New Jersey, California, Louisiana, New York, Massachusetts and Texas, where majority of these at-risk facilities are located) and analyze the location of hazardous sites (e.g., coastal power plants, sewage treatment facilities, fossil fuel infrastructure, industrial facilities, and formerly used defense sites) that are likely to be impacted by a 1-in-100-year flood event by 2100.

They find that after controlling for population density and county level factors, the Hispanic, poor, non-voters and renters face disproportionately high risk of exposure. The authors discuss the environmental justice implications of increasing level of this risk exposure due to the location of hazardous waste sites located in low-lying coastal area.

While the authors put significant effort in conducting the analysis, it is primarily based on correlation rather than causation. Some attempts to establish the direction of causal relationship (between the extent of risk exposure and dwelling locations of socially marginalized population) can significantly improve the contribution of the paper.

For instance, it is important to isolate to what extent marginalized populations living in the areas are exposed this emerging risk because of the siting of these facilities. And to what extent they are accepting these risks as compensatory variations of other benefits/amenities (e.g. higher wage and lower cost of housing).

(Remarks on code availability)

Version 1:

Reviewer comments:

Reviewer #1

(Remarks to the Author)

Dear Authors, thank you for your thorough response to reviewer document and for the work you did to improve your manuscript. Based on my second review this is now ready for publication. It is an excellent research paper that is well written and timely.

(Remarks on code availability)

Reviewer #2

(Remarks to the Author)

I have reviewed the paper. The response from authors considers my concerns as 'outside the purview of the paper'.

There are ways to exploit variations in land use, wage, permitting and siting related characteristics to establish causal relations (and isolate to what extent it is because of the siting of these facilities vs. to what extent it is due to compensatory variations of other benefits/amenities).

I think the paper should not be considered without accounting for these concerns.

(Remarks on code availability)

RESPONSE TO REFEREES

We appreciate the reviewers for their thoughtful comments which were helpful in strengthening the paper.

Reviewer #1

COMMENT: This is an interesting paper and I appreciate the hard work you put into to get to this point. You have provide a rigorous analysis of hazardous sites at risk from future flooding caused by sea level rise. They focus on the types of communities impacted by these exposures and bring to light the significant disparities. The data and methods for this paper are technically sound and well documented. They make some excellent points and compare their work with the current state of the literature. They make some very good points, some of which I note below.

RESPONSE: Thank you for the positive feedback.

In my opinion, there are also some shortcomings that should be addressed to allow the paper to move the field forward.

RESPONSE: Our responses are outline below. Line numbers refer to the manuscript when viewed without tracked changes.

COMMENT: This paper focuses exclusively on the United States. It is very well done and leverages important and relevant data sets. It is also timely and relevant as others have worked to call attention to this threat. However, it does not connect its findings and work to a global audience. I think the paper would be stronger if there was a thread that connected the US context with other countries.

RESPONSE:

We agree with the reviewer's comment on the importance of connecting our work to global contexts. We have added text to emphasize the global importance of the issue of SLR and industrial sites to our introduction (lines 54-58) and have now added reference to studies in other global regions to our Discussion section (lines 255-258) to highlight relevant work on SLR risks to infrastructure in non-US contexts.

“Around the world, industrial facilities are disproportionately located along coastlines due to the historical importance of maritime trade to the establishment of industrial port cities, strategic access to global trade routes for raw materials and finished products via ports, and need for sea water for cooling and wastewater disposal. “

“Studies outside of the U.S., for example in coastal regions in India, Copenhagen, Vietnam and Italy have investigated the risks posed by climate change-driven SLR and storm surge on infrastructure and other vulnerable sites, but none to our knowledge have evaluated these risks using an environmental justice framework.”

COMMENT: The methods are predominantly clear, articulate, and rigorous. The major issue I have with the methods is that there are no spatial visualizations in the supplementary material to help people understand their process. One example is their use of Euclidean distance. It's my

understanding that they measured 3 kilometers towards shore from the 10 meter above mean high tide line to find low-lying areas. If that's true that method has flaws since those locations could be higher in elevation depending on the local topography. Maps explaining these methods would greatly improve their methods section.

RESPONSE: We added two schematics to the supplemental material that illustrate the methods used to define the study area and at-risk block groups (Figure S5) and estimate flood risk (Figure S6). We agree the term "low-lying" was confusing. We replaced it with "coastal" throughout the text and clarified in the methods that Euclidean distance was used solely to exclude inland areas not at conceivable risk of flooding due to sea level rise from further consideration (lines 317-320):

*"The spatial extent of our analysis was the land area within a 3-kilometer Euclidean distance of the 10 meters elevation above current mean high tide across all coastal U.S. states and Puerto Rico (see **Supplemental Figure S5**). Areas farther inland are at no conceivable risk of flooding due to sea level rise this century and were therefore excluded."*

Euclidean distance was not utilized in the estimation of flood risk. Flood risk was estimated using probabilistic sea level rise projections, site topographic information and connectivity to the ocean, and coastal flood height return levels from historical tide station data as described in lines 372-385 and illustrated in the new Figure S6.

COMMENT: Another concern was their use of CJEST. The authors mentioned it in the discussion and in their figure 2. However, there is no mention of it in their methods. This dataset should be more clearly defined and explained. Since the dataset is a combination of the other variables they are using it should be more thoroughly scrutinized and then explained.

RESPONSE: Thank you for catching this omission in the Method section. In response we have added the following text explaining the CEJST in the Methods section (Lines 400-408):

"Finally, we used the federal Climate and Economic Justice Screening Tool (CEJST) that identifies disadvantaged communities in all 50 states, the District of Columbia, and U.S. Territories⁵⁵. Developed as part of the Justice40 Initiative, CEJST was used by federal agencies to identify disadvantaged communities facing disproportionate climate and environmental burdens as well as economic marginalization. CEJST identifies disadvantaged communities through eight categories of vulnerability metrics related to climate change, energy, health, housing, legacy pollution, transportation, water and wastewater, and workforce development. Census tracts are identified as disadvantaged if they meet 90th percentile thresholds for indicators within any of the eight categories and are at or above the 65th percentile for low-income."

We note that the CEJST was recently removed by the Trump Administration from the website of the White House Council on Environmental Quality, but this tool and meta data are now posted on a website at Harvard University, which we cite here:

White House Council on Environmental Quality: Climate and Economic Justice Screening Tool Technical Support Document, Version 2.0. December 2024. Available at: <https://dataverse.harvard.edu/dataset.xhtml?persistentId=doi:10.7910/DVN/ZVKXVQ>

COMMENT: Their results are a meaningful contribution to the field and bring to light the significant risk to vulnerable populations from hazardous waste sites at risk of sea level rise. The data they present supports this conclusion. Their findings that 70% of the sites are in seven states also gives policy makers and advocates an area of focus. The subsequent recommendation of making sure information of these risks is available seems narrow.

RESPONSE: In response to this comment, we have added the following text in the conclusion of the paper (Lines 299-305)

“Risks may be reduced through enhanced regulatory requirements (1) for at-risk facilities to mitigate and prevent contamination threats and (2) for more robust assessment of clean-up sites to inform abatement activities and decisions about future land reuse. Action-oriented partnerships between communities living near at-risk sites and government agencies at local, state, and federal levels may increase the chances for success of these strategies by marshalling much-needed resources aimed at preventing contamination from acute natech disasters and slower-moving threats, including SLR-related flooding. “

COMMENT: The results from this paper are interesting and possibly significant. It would help if they further contextualized their findings relative to other hazards. Is sea-level rise the greatest threat to hazardous waste sites? Looking at the Summers, K., Lamper, A., & Buck, K. (2021) paper would help to support their work.

RESPONSE: We thank the reviewer for sharing this reference, which we have added to the paper in lines 296-299, where we point out that SLR-related flooding is one of myriad climate-related and natural events that can potentially threaten clean-up sites as well as facilities that use or store hazardous material.

“Many other climate-related phenomena, such as groundwater rise, wildfires, landslides, major storms, and extreme heat, also threaten clean-up sites and active facilities that use and store hazardous materials. To achieve a fuller picture, information on these threats should be integrated with projected SLR flood risk data.”

Reviewer #2:

COMMENT: The paper focuses on a very important topic and attempts to analyze the extent of exposure of SLR-related flooding of hazardous sites to the socially marginalized people in the USA. The authors focus on seven states in the USA (Florida, New Jersey, California, Louisiana, New York, Massachusetts and Texas, where majority of these at-risk facilities are located) and analyze the location of hazardous sites (e.g., coastal power plants, sewage treatment facilities, fossil fuel infrastructure, industrial facilities, and formerly used defense sites) that are likely to be impacted by a 1-in-100-year flood event by 2100.

They find that after controlling for population density and county level factors, the Hispanic, poor, non-voters and renters face disproportionately high risk of exposure. The authors discuss the environmental justice implications of increasing level of this risk exposure due to the location of hazardous waste sites located in low-lying coastal area.

While the authors put significant effort in conducting the analysis, it is primarily based on correlation rather than causation. Some attempts to establish the direction of causal relationship (between the extent of risk exposure and dwelling locations of socially marginalized population) can significantly improve the contribution of the paper.

For instance, it is important to isolate to what extent marginalized populations living in the areas are exposed this emerging risk because of the siting of these facilities. And to what extent they are accepting these risks as compensatory variations of other benefits/amenities (e.g. higher wage and lower cost of housing).

RESPONSE: The reviewer poses a key question regarding the extent to which marginalized populations living near at-risk sites are there because of prior siting decisions or whether they have in some way chosen to live in these locations due to other factors. This encapsulates a much studied “Which came first?” discussion in the EJ literature. Although our census data preclude the ability to evaluate what key factors have driven the residential location decisions of the marginalized groups living near the at-risk sites that we identified in our analysis and an in-depth assessment of “Which came first?” is beyond the purview of this paper, we have added reference to this body of literature in our introduction (Lines 60-62).

“Moreover, longitudinal analyses show that disproportionate hazard burdens faced by racially and economically marginalized groups are largely due to legacies of discriminatory land-use, permitting, and facility siting decisions.”

RESPONSE TO REFEREES

Thank you for this second round of reviews to which we respond below.

Reviewer #1: Dear Authors, thank you for your thorough response to reviewer document and for the work you did to improve your manuscript. Based on my second review this is now ready for publication. It is an excellent research paper that is well written and timely.

RESPONSE: We thank the reviewer for their positive feedback.

Reviewer #2: I have reviewed the paper. The response from authors considers my concerns as ‘outside the purview of the paper’.

There are ways to exploit variations in land use, wage, permitting and siting related characteristics to establish causal relations (and isolate to what extent it is because of the siting of these facilities vs. to what extent it is due to compensatory variations of other benefits/amenities).

I think the paper should not be considered without accounting for these concerns.

RESPONSE: We appreciate the reviewer’s reiteration of their concerns. However, undertaking the kind of analysis that the reviewer urges we do would require a completely different hypothesis and new line of inquiry; for example, an assessment of the so-called “minority move-in versus facility siting hypothesis” would interrogate “Which came first? The siting of facilities at-risk, or minorities moving into/White people moving out of neighborhoods driven by other factors, including temporal shifts in land use decision-making, wages, industrial development, job opportunities, the housing market, etc.?” We agree such a comprehensive temporal analysis of multiple variables would be worthwhile, but it would require several sources of critical data to which we do not have access at a national scale. Indeed, we would need information on when each of the nearly 48,000 facilities included in our analysis were originally sited (which is not available in our dataset), along with time-varying data (pre-/post facility siting years) on changes in local population demographics (which are not available for all racial/ethnic groups in our analysis due to changes in the phrasing of US Census race and ethnicity questions over time). We would additionally require localized (municipal and/or regional) information on land use and zoning changes, industrial development trends in key economic sectors related to our 11 facility categories, as well as shifts in wages, employment, infrastructure, housing markets, housing and transportation development in order to understand the causes for any shifts in demographics and population changes over time. While this type of deep, multivariable temporal analysis might be feasible at a very localized scale in a few municipalities where such data might be painstakingly amassed, it is currently not feasible to collect such a broad scope of temporal variables to support the kind of analysis that this reviewer is urging us to do at a national scale.

Nevertheless, although we did not undertake the wholesale analysis proposed by the reviewer, in response to their comment we have further elevated the importance of their question as an important future line of inquiry that might be feasible at a much narrower geographic scope, and with a focus on a smaller subset of facilities. In our response to this comment by the Reviewer in the first round of comments, we had added the following text in the Introduction, which remains:

“...longitudinal analyses show that disproportionate hazard burdens faced by racially and economically marginalized groups are largely due to discriminatory land-use, permitting, and facility siting decisions⁸⁻¹¹. “

To address the reviewer’s continuing concern about this issue in their second round of feedback, we have added new text in the Discussion to emphasize need for future research to address this question:

“Future research focusing on a smaller subset of facilities and more localized regions or municipalities could further elucidate and potentially untangle the extent to which place-based trends in industrial, economic, labor market and housing development trajectories, demographic churning, changes in land-use decision-making as well as other shifting structural factors account for the origins and persistence of inequities in exposures to at-risk sites that disproportionately affect marginalized populations.”

Additional changes we made to this revised manuscript:

We also wanted to highlight the following changes we made in this revised version of the manuscript.

- 1) We corrected the text to clarify that the spatial extent of our analysis was U.S counties and county equivalents with any land area below 18 meters elevation above current mean high tide across all coastal U.S. states and Puerto Rico
- 2) For consistency with our prior published work, we then defined the universe of included facilities in our analysis as all sites within these counties In the prior version of manuscript, we had restricted this initial universe of facilities to only those near populated areas. However upon further reflection we felt it important to include facilities not near populated area 1) for comparability with prior work and 2) since the 1km distance we used to define “near” is somewhat arbitrary and the distance that contaminants travel can vary, with some releases such as those from major oil spills often impacting populations much farther away.
- 3) For oil and gas wells, we continue to use latitude and longitude point locations to represent sites due to the small size of well pads compared to other site categories. However, in this revised manuscript we sought to eliminate offshore oil and gas wells by identifying those wells that were beyond the boundaries of 2010 Census block groups and excluding them from the analysis to focus on wells located on land. To do this, we used block group boundaries from the National Historical Geographic Information System, which do not include coastal water areas and terminate at the coastline.

- 4) These changes have resulted in an increase in total number of facilities in our analysis from 32,239 to 47,646 (See revised Table 1), with the largest increase in active oil and gas wells (from 10,599 to 24,095). This also changed the results in Figure 1, in which oil and gas wells now figure much more predominantly in the proportion of at-risk sites, in particular for Louisiana and Texas. However, the same top 7 states still account for the overwhelming proportion (nearly 80%) of the at-risk facilities.
- 5) For our statistical analysis examining demographic factors associated with proximity to at-risk facilities or flooding severity, the universe of block groups examined remains unchanged (e.g., Table 2, and Figures 3 and 4). After re-running our models effect estimates were also very similar.
- 6) We have added another co-author to this revised manuscript, Alique Berberian in light of her help with data curation and edits on this second round of revisions. The signed approvals of this author change have been provided by all of the original co-authors.